# Max-MIG: an Information Theoretic Approach for Joint Learning from Crowds

**Peng Cao**[*]**, Yilun Xu**[*]
School of Electronics Engineering and Computer Science,
Peking University
{caopeng2016,xuyilun}@pku.edu.cn

**Yuqing Kong**
The Center on Frontiers of Computing Studies,
Peking University
yuqing.kong@pku.edu.cn

**Yizhou Wang**
Nat'l Eng. Lab. for Video Technology
Computer Science Dept., Peking University
Cooperative Medianet Innovation Center
PengCheng Lab
Deepwise AI Lab
Yizhou.Wang@pku.edu.cn

## Abstract

Eliciting labels from crowds is a potential way to obtain large labeled data. Despite a variety of methods developed for learning from crowds, a key challenge remains unsolved: *learning from crowds without knowing the information structure among the crowds a priori, when some people of the crowds make highly correlated mistakes and some of them label effortlessly (e.g. randomly).* We propose an information theoretic approach, Max-MIG, for joint learning from crowds, with a common assumption: the crowdsourced labels and the data are independent conditioning on the ground truth. Max-MIG simultaneously aggregates the crowdsourced labels and learns an accurate data classifier. Furthermore, we devise an accurate data-crowds forecaster that employs both the data and the crowdsourced labels to forecast the ground truth. To the best of our knowledge, this is the first algorithm that solves the aforementioned challenge of learning from crowds. In addition to the theoretical validation, we also empirically show that our algorithm achieves the new state-of-the-art results in most settings, including the real-world data, and is the first algorithm that is robust to various information structures.

## 1 Introduction

Lack of large labeled data is a notorious bottleneck of the data-driven-based machine learning paradigm. Crowdsourcing provides a potential solution to this challenge: eliciting labels from crowds. However, the elicited labels are usually very noisy, especially for some difficult tasks (e.g. age estimation, medical images annotation). In the crowdsourcing-learning scenario, two problems are raised:

(i) *how to aggregate and infer the ground truth from the imperfect crowdsourced labels?*

(ii) *how to learn an accurate data classifier with the imperfect crowdsourced labels?*

One conventional solution to the two problems is aggregating the crowdsourced labels using majority vote and then learning a data classifier with the majority answer. However, this naive method will cause biased results when the task is difficult and the majority of the crowds label randomly or always label a particular class (say class 1) effortlessly.

Another typical solution is aggregating the crowdsourced labels in a more clever way, like spectral method (Dalvi et al., 2013; Zhang et al., 2014), and then learning with the aggregated results. This method avoids the above flaw that the majority vote method has, as long as their randomnesses are

---

[*]Equal Contribution.

mutually independent. However, the spectral method requires that the experts' labeling noise are mutually independent, which often does not hold in practice since some experts may make highly correlated mistakes (see Figure 2 for example). Moreover, the above solutions aim to train an accurate data classifier and do not provide a method that can employ both the data and the crowdsourced labels to forecast the ground truth.

A common assumption in the learning from crowds literature is that conditioning on the ground truth, the crowdsourced labels and the data are independent, as shown in Figure 1 (a). Under this assumption, the crowdsourced labels correlate with the data due to and *only* due to the ground truth. Thus, this assumption tells us the ground truth is the "information intersection" between the crowdsourced labels and the data. This "information intersection" assumption does not restrict the information structure among the crowds i.e. this assumption still holds even if some people of the crowds make highly correlated mistakes.

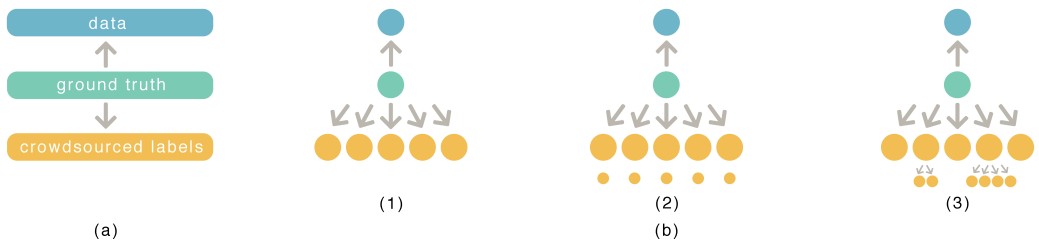

Figure 1: (a) The general information structure under the "information intersection" assumption. (b) Possible information structures under the "information intersection" assumption, where the crowdsourced labels are provided by several experts: (1) *independent mistakes:* all of the experts are correlated with the ground truth and mutually independent of each other conditioning on the ground truth; for (2), (3) the senior experts are mutually conditional independent and (2) *naive majority:* the junior experts always label class 1 without any effort; (3) *correlated mistakes:* the junior experts, who were advised by the same senior expert before, make highly correlated mistakes.

We present several possible information structures under the "information intersection" assumption in Figure 1 (b). The majority vote will lead to inaccurate results in all cases if the experts have different levels of expertise and will induce extremely biased results in case (2) when a large number of junior experts always label class 1. The approaches that require the experts to make independent mistakes will lead to biased results in case (3), when the experts make highly correlated mistakes

In this paper, we propose an information theoretic approach, Max-MIG, for joint learning from crowds, with a common assumption: the crowdsourced labels and the data are independent conditioning on the ground truth. *To the best of our knowledge, this is the first algorithm that is both theoretically and empirically robust to the situation where some experts make highly correlated mistakes and some experts label effortlessly, without knowing the information structure among the experts.* Our algorithm simultaneously aggregates the crowdsourced labels and learns an accurate data classifier. In addition, we propose a method to learn an accurate data-crowds forecaster that can employ both the data and the crowdsourced labels.

At a high level, our algorithm trains a data classifier and a crowds aggregator simultaneously to maximize their "mutual information". This process will find the "information intersection" between the data and crowdsourced labels i.e. the ground truth labels. The data-crowds forecaster can be easily constructed from the trained data classifier and the trained crowds aggregator. This algorithm allows the conditional dependency among the experts as long as the intersection assumption holds.

We design the crowds aggregator as the "weighted average" of the experts. This simple "weighted average" form allows our algorithm to be both highly efficient in computing and theoretically robust to a large family of information structures (e.g. case (1), (2), (3) in Figure 1 (b)). Particularly, our algorithm works when there exists a subset of senior experts, whose identities are *unknown*, such that these senior experts have mutually independent labeling biases and it is sufficient to only use the seniors' information to predict the ground truth label. For other junior experts, they are allowed to have any dependency structure among themselves or between them and the senior experts.

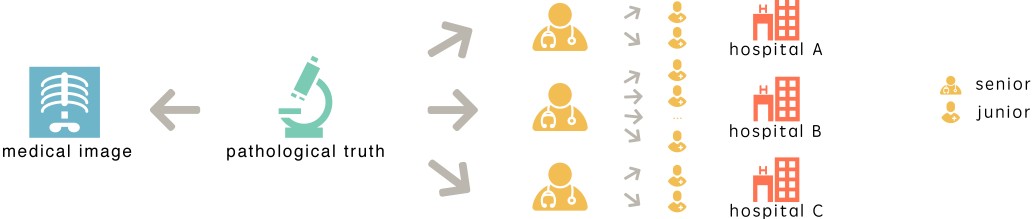

Figure 2: Medical image labeling example: we want to train a data classifier to classify the medical images into two classes: benign and malignant. Each image is labeled by several experts. The experts are from different hospitals, say hospital A, B, C. Each hospital has a senior who has a high expertise. We assume the seniors' labeling biases are mutually independent. However, for two juniors that were advised by the same senior before, they make highly correlated mistakes when labeling the images. We assume that 5 experts are from hospital A, 50 experts are from hospital B, and 5 experts are from hospital C. If we use majority vote to aggregate the labels, the aggregated result will be biased to hospital B. If we still pretend the experts' labeling noises are independent and apply the approaches that require independent mistakes, the aggregated result will still be biased to hospital B.

## 2  RELATED WORK

A series of works consider the learning from crowds problem and mix the learning process and the aggregation process together. Raykar et al. (2010) reduce the learning from crowds problem to a maximum likelihood estimation (MLE) problem, and implement an EM algorithm to jointly learn the expertise of different experts and the parameters of a logistic regression classifier. Albarqouni et al. (2016) extend this method to combine with the deep learning model. Khetan et al. (2017) also reduce the learning problem to MLE and assume that the optimal classifier gives the ground truth labels and the experts make independent mistakes conditioning on the ground truth. Unlike our method, these MLE based algorithms are not robust to correlated mistakes. Recently, Guan et al. (2017) and Rodrigues & Pereira (2017) propose methods that model multiple experts individually and explicitly in a neural network. However, their works lack theoretical guarantees and are outperformed by our method in the experiments, especially in the naive majority case. Moreover, unlike our method, their methods cannot be used to employ both the data and the crowdsourced labels to forecast the ground truth.

Several works focus on modeling the experts. Whitehill et al. (2009) model both expert competence and image difficulty, but did not consider expert bias. Welinder et al. (2010) model each expert as a multidimensional classifier in an abstract feature space and consider both the bias of the expert and the difficulty of the image. Rodrigues et al. (2014) model the crowds by a Gaussian process. Khetan & Oh (2016); Shah et al. (2016) consider the generalized Dawid-Skene model (Dawid & Skene, 1979) which involves the task difficulty. However, these works are still not robust to correlated mistakes. We model the crowds via the original Dawid-Skene model and do not consider the task difficulty, but we believe our Max-MIG framework can be incorporated with any model of the experts and allow correlated mistakes.

Our method differs from the works that focus on inferring ground truth answers from the crowds' reports and then learn the classifier with the inferred ground truth (e.g. (Dawid & Skene, 1979; Zhou et al., 2012; Liu et al., 2012; Karger et al., 2014; Zhang et al., 2014; Dalvi et al., 2013; Ratner et al., 2016)) since our method simultaneously infers the ground truth and learns the classifier. In addition, our method provides a data-crowds forecaster while those works do not.

Our method is also closely related to co-training. Blum & Mitchell (1998) first propose the co-training framework: simultaneously training two classifiers to aggregate two views of data. Our method interprets joint learning from crowds as a co-training style problem. Most traditional co-training methods require weakly good classifier candidates (e.g. better than random guessing). We follow the general information theoretic framework proposed by Kong & Schoenebeck (2018) that does not have this requirement. However, Kong & Schoenebeck (2018) only provide theoretic framework and assume an extremely high model complexity without considering the over-fitting issue, which is a too

strong assumption for practice. Our work apply this framework to the learning from crowds problem and provide the proper design for the model complexity as well as the experimental validations.

## 3 METHOD

In this section, we formally define the problem, introduce our method, Max-MIG, and provide a theoretical validation for our method.

**Notations** For every set $\mathcal{A}$, we use $\Delta_{\mathcal{A}}$ to denote the set of all possible distributions over $\mathcal{A}$. For every integer $M$, we use $[M]$ to denote $\{1, 2, \ldots, M\}$. For every matrix $\mathbf{A} = (A_{i,j})_{i,j} \in \mathbb{R}^{+s \times t}$, we define $\log \mathbf{A}$ as a $s \times t$ matrix such that its the $(i,j)^{th}$ entry is $\log(A_{i,j})$. Similarly for every vector $\mathbf{v} = (v_i)_i \in \mathbb{R}^{+s}$, we define $\log \mathbf{v}$ as a vector such that its the $i^{th}$ entry is $\log(v_i)$.

**Problem statement** There are $N$ datapoints. Each datapoint $x \in I$ (e.g. the CT scan of a lung nodule) is labeled by $M$ experts $y^{[M]} := \{y^1, y^2, \ldots, y^M | y^m \in \mathcal{C}\}$ (e.g. $\mathcal{C} = \{\text{benign}, \text{malignant}\}$, 5 experts' labels: {benign, malignant, benign, benign, benign}). The datapoint $x$ and the crowdsourced labels $y^{[M]}$ are related to a ground truth $y \in \mathcal{C}$ (e.g. the pathological truth of the lung nodule).

We are aiming to simultaneously train a data classifier $h$ and a crowds aggregator $g$ such that $h : I \mapsto \Delta_{\mathcal{C}}$ predicts the ground truth $y$ based on the datapoint $x \in I$, and $g : \mathcal{C}^M \to \Delta_{\mathcal{C}}$ aggregates $M$ crowdsourced labels $y^{[M]}$ into a prediction for ground truth $y$. We also want to learn a data-crowds forecaster $\zeta : I \times \mathcal{C}^M \mapsto \Delta_{\mathcal{C}}$ that forecasts the ground truth $y$ based on both the datapoint $x \in I$ and the crowdsourced labels $y^{[M]} \in \mathcal{C}$.

### 3.1 MAX-MIG: AN INFORMATION THEORETIC APPROACH

Figure 3 illustrates the overview idea of our method. Here we formally introduce the building blocks of our method.

**Data classifier $h$** The data classifier $h$ is a neural network with parameters $\Theta$. Its input is a datapoint $x$ and its output is a distribution over $\mathcal{C}$. We denote the set of all such data classifers by $H_{NN}$.

**Crowds aggregator $g$** The crowds aggregator $g$ is a "weighted average" function to aggregate crowdsourced labels with parameters $\{\mathbf{W}^m \in \mathbb{R}^{|\mathcal{C}| \times |\mathcal{C}|}\}_{m=1}^M$ and $\mathbf{b}$. Its input $y^{[M]}$ is the crowdsourced labels provided by $M$ experts for a datapoint and its output is a distribution over $\mathcal{C}$. By representing each $y^m \in y^{[M]}$ as an one-hot vector $\mathbf{e}^{(y^m)} := (0, \ldots, 1, \ldots, 0)^\top \in \{0,1\}^{|\mathcal{C}|}$ where only the $y^m$th entry of $\mathbf{e}^{(y^m)}$ is 1,

$$g(y^{[M]}; \{\mathbf{W}^m\}_{m=1}^M, \mathbf{b}) = \text{softmax}\left(\sum_{m=1}^M \mathbf{W}^m \cdot \mathbf{e}^{(y^m)} + \mathbf{b}\right)$$

where $\mathbf{W}^m \cdot \mathbf{e}^{(y^m)}$ is equivalent to pick the $y^m$th column of matrix $\mathbf{W}^m$, as shown in Figure 3. We denote the set of all such crowds aggregators by $G_{WA}$.

**Data-crowds forecaster $\zeta$** Given a data classifier $h$, a crowds aggregator $g$ and a distribution $\mathbf{p} = (p_c)_c \in \Delta_{\mathcal{C}}$ over the classes, the data-crowds forecaster $\zeta$, that forecasts the ground truth based on both the datapoint $x$ and the crowdsourced labels $y^{[M]}$, is constructed by

$$\zeta(x, y^{[M]}; h, g, \mathbf{p}) = \text{Normalize}\left(\left(\frac{h(x)_c \cdot g(y^{[M]})_c}{p_c}\right)_c\right)$$

where $\text{Normalize}(\mathbf{v}) := \frac{\mathbf{v}}{\sum_c v_c}$.

**$f$-mutual information gain $MIG^f$** $f$-mutual information gain $MIG^f$ measures the "mutual information" between two hypotheses, which is proposed by Kong & Schoenebeck (2018). Given $N$ datapoints $x_1, x_2, \ldots, x_N \in I$ where each datapoint $x_i$ is labeled by $M$ crowdsourced labels

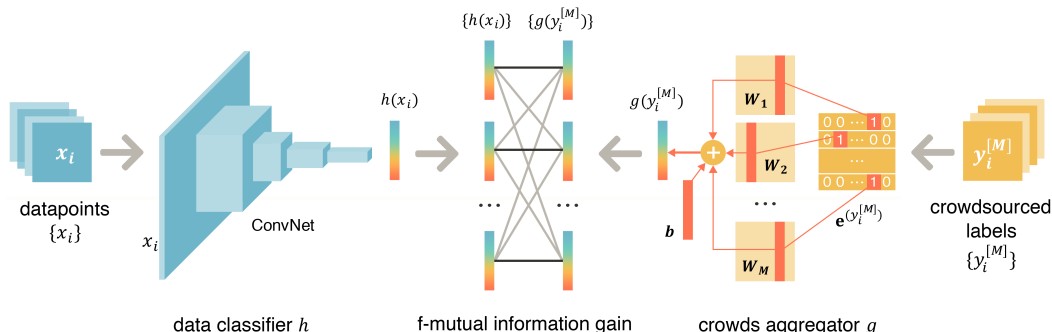

Figure 3: Max-MIG overview: *Step 1: finding the "information intersection" between the data and the crowdsourced labels*: we train a data classifier $h$ and a crowds aggregator $g$ simultaneously to maximize their $f$-mutual information gain $MIG^f(h, g, \mathbf{p})$ with a hyperparameter $\mathbf{p} \in \Delta_{\mathcal{C}}$. $h$ maps each datapoint $x_i$ to a forecast $h(x_i) \in \Delta_{\mathcal{C}}$ for the ground truth. $g$ aggregates $M$ crowdsourced labels $y_i^{[M]}$ into a forecast $g(y_i^{[M]}) \in \Delta_{\mathcal{C}}$ by "weighted average". We tune the parameters of $h$ and $g$ simultaneously to maximize their $f$-mutual information gain. We will show the maximum is the $f$-mutual information (a natural extension of mutual information, see Appendix C) between the data and the crowdsourced labels. *Step 2: aggregating the "information intersection"*: after we obtain the best $h, g, \mathbf{p}$ that maximizes $MIG^f(h, g, \mathbf{p})$, we use them to construct a data-crowds forecaster $\zeta$ that forecasts ground truth based on both the datapoint and the crowdsourced labels.

To calculate the $f$-mutual information gain, we reward them for the average "agreements" between their outputs for the *same* task, i.e. $h(x_i)$ and $g(y_i^{[M]})$ , as shown by the black lines, and punish them for the average "agreements" between their outputs for the *different* tasks, i.e. $h(x_i)$ and $g(y_j^{[M]})$ where $i \neq j$, as shown by the grey lines. Intuitively, the reward encourages the data classifier to agree with the crowds aggregator, while the punishment avoids them naively agreeing with each other, that is, both of them map everything to $(1, 0, \dots, 0)$. The measurement of "agreement" depends on the selection of $f$. See formal definition for $MIG^f$ in (1).

$y_i^1, y_i^2, \dots, y_i^M \in \mathcal{C}$, the $f$-mutual information gain between $h$ and $g$, associated with a hyperparameter $\mathbf{p} = (p_c)_c \in \Delta_{\mathcal{C}}$, is defined as the average "agreements" between $h$ and $g$ for the same task minus the average "agreements" between $h$ and $g$ for the different tasks, that is,

$$MIG^f(\{x_i\}, \{y_i^{[M]}\}; h, g, \mathbf{p}) = \frac{1}{N} \sum_i \partial f \left( \sum_{c \in \mathcal{C}} \frac{h(x_i)_c \cdot g(y_i^{[M]})_c}{p_c} \right)$$
$$- \frac{1}{N(N-1)} \sum_{i \neq j} f^\star \left( \partial f \left( \sum_{c \in \mathcal{C}} \frac{h(x_i)_c \cdot g(y_j^{[M]})_c}{p_c} \right) \right) \tag{1}$$

where $f$ is a convex function satisfying $f(1) = 0$ and $f^\star$ is the Fenchel duality of $f$. We can use Table 1 as reference for $\partial f(\cdot)$ and $f^\star(\partial f(\cdot))$.

Table 1: Reference for common $f$-divergences and corresponding $MIG^f$'s building blocks. This table is induced from Nowozin et al. (2016).

| $f$-divergence | $f(t)$ | $\partial f(K)$ | $f^\star(\partial f(K))$ |
|---|---|---|---|
| KL divergence | $t \log t$ | $1 + \log K$ | $K$ |
| Pearson $\chi^2$ | $(t-1)^2$ | $2(K-1)$ | $K^2 - 1$ |
| Jensen-Shannon | $-(t+1) \log \frac{t+1}{2} + t \log t$ | $\log \frac{2K}{1+K}$ | $-\log(\frac{2}{1+K})$ |

Since the parameters of $h$ is $\Theta$ and the parameters of $g$ is $\{\mathbf{W}^m\}_{m=1}^M$ and $\mathbf{b}$, we naturally rewrite $MIG^f(\{x_i\}, \{y_i^{[M]}\}; h, g, \mathbf{p})$ as

$$MIG^f(\{x_i\}, \{y_i^{[M]}\}; \Theta, \{\mathbf{W}^m\}_{m=1}^M, \mathbf{b}, \mathbf{p}).$$

We seek $\{\Theta, \{\mathbf{W}^m\}_{m=1}^M, \mathbf{b}, \mathbf{p}\}$ that maximizes $MIG^f$. Later we will show that when the prior of the ground truth is $\mathbf{p}^*$ (e.g. $\mathbf{p}^* = (0.8, 0.2)$ i.e. the ground truth is benign with probability 0.8 and malignant with probability 0.2 a priori), the best $\mathbf{b}$ and $\mathbf{p}$ are $\log \mathbf{p}^*$ and $\mathbf{p}^*$ respectively. Thus, we can set $\mathbf{b}$ as $\log \mathbf{p}$ and only tune $\mathbf{p}$. When we have side information about the prior $\mathbf{p}^*$, we can fix parameter $\mathbf{p}$ as $\mathbf{p}^*$, and fix parameter $\mathbf{b}$ as $\log \mathbf{p}^*$.

## 3.2 THEORETICAL JUSTIFICATION

This section provides a theoretical validation for Max-MIG, i.e., maximizing the $f$-mutual information gain over $H_{NN}$ and $G_{WA}$ finds the "information intersection" between the data and the crowdsourced labels. In Appendix E, we compare our method with the MLE method (Raykar et al., 2010) theoretically and show that unlike our method, MLE is not robust to the correlated mistakes case.

Recall that we assume that conditioning on the ground truth, the data and the crowdsourced labels are mutually independent. Thus, we can naturally define the "information intersection" as a pair of data classifier and crowds aggregator $h^*, g^*$ such that they both fully use their input to forecast the ground truth. Kong & Schoenebeck (2018) shows that when we have infinite number of datapoints and maximize over all possible data classifiers and crowds aggregators, the "information intersection" will maximize $MIG^f(h, g)$ to the $f$-mutual information (Appendix C) between the data and the crowdsourced labels. However, in practice, with a finite number of datapoints, the data classifier and the crowds aggregator space should be not only sufficiently rich to contain the "information intersection" but also sufficiently simple to avoid over-fitting. Later, the experiment section will show that our picked $H_{NN}$ and $G_{WA}$ are sufficiently simple to avoid over-fitting. We assume the neural network space is sufficiently rich. It remains to show that our weighted average aggregator space $G_{WA}$ is sufficiently rich to contain $g^*$.

**Model and assumptions** Each datapoint $x_i$ with crowdsourced labels provided by $M$ experts $y_i^1, ..., y_i^M$ are drawn i.i.d. from random variables $X, Y^1, ..., Y^M$.

**Assumption 3.1** (Co-training assumption). *$X$ and $Y^{[M]}$ are independent conditioning on $Y$.*

Note that we do not assume that the experts' labels are conditionally mutually independent. We define $\mathbf{p}^* \in \Delta_{\mathcal{C}}$ as the prior for $Y$, i.e. $p_c^* = P(Y = c)$.

**Definition 3.2** (Information intersection). *We define $h^*$, $g^*$ and $\zeta^*$ such that*

$$h^*(x)_c = P(Y = c | X = x) \quad g^*(y^{[M]})_c = P(Y = c | Y^{[M]} = y^{[M]}).$$

$$\zeta^*(x, y^{[M]})_c = P(Y = c | X = x, Y^{[M]} = y^{[M]})$$

*We call them Bayesian posterior data classifier / crowds aggregator / data-crowds forecaster respectively. We call $(h^*, g^*)$ the information intersection between the data and the crowdsourced labels.*

We also assume the neural network space is sufficiently rich to contain $h^*$.

**Assumption 3.3** (Richness of the neural networks). *$h^* \in H_{NN}$.*

**Theorem 3.4.** *With assumptions 3.1, 3.3, when there exists a subset of experts $\mathcal{S} \subset [M]$ such that the experts in $\mathcal{S}$ are mutually independent conditioning on $Y$ and $Y^{\mathcal{S}}$ is a sufficient statistic for $Y$, i.e. $P(Y = y | Y^{[M]} = y^{[M]}) = P(Y = y | Y^{\mathcal{S}} = y^{\mathcal{S}})$ for every $y \in \mathcal{C}, y^{[M]} \in \mathcal{C}^M$, then $(h^*, g^*, \mathbf{p}^*)$ is a maximizer of*

$$\max_{h \in H_{NN}, g \in G_{WA}, \mathbf{p} \in \Delta_{\mathcal{C}}} \mathbb{E}_{X, Y^{[M]}} MIG^f(h(X), g(Y^{[M]}), \mathbf{p})$$

*and the maximum is the $f$-mutual information between $X$ and $Y^{[M]}$. Moreover, $\zeta^*(x, y^{[M]}) = \zeta(x, y^{[M]}; h^*, g^*, \mathbf{p}^*)$ for every $x, y^{[M]}$.*

Our main theorem shows that if there exists a subset of senior experts such that these senior experts are mutually conditional independent and it is sufficient to only use the information from these senior experts, then Max-MIG finds the "information interstion". Note that we do not need to know the identities of the senior experts. For other junior experts, we allow any dependency structure among them and between them and the senior experts. Moreover, this theorem also shows that our method handles the independent mistakes case where all experts can be seen as senior experts (Proposition D.3).

To show our results, we need to show that $G_{WA}$ contains $g^*$, i.e. there *exists* proper weights such that $g^*$ can be represented as a weighted average. In the independent mistakes case, we can construct each expert's weight using her confusion matrix. Thus, in this case, each expert's weight represents her expertise. In the general case, we can construct each senior expert's weight using her confusion matrix and make the junior experts' weights zero. Due to space limitation, we defer the formal proofs to Appendix D.

## 4 EXPERIMENT

In this section, we evaluate our method on image classification tasks with both synthesized crowd-sourced labels in various of settings and real world data.

Our method **Max-MIG** is compared with: **Majority Vote**, training the network with the major vote labels from all the experts; **Crowd Layer**, the method proposed by Rodrigues & Pereira (2017); **Doctor Net**, the method proposed by Guan et al. (2017) and **AggNet**, the method proposed by Albarqouni et al. (2016).

**Image datasets**    Three datasets are used in our experiments. The **Dogs vs. Cats** (Kaggle, 2013) dataset consists of $25,000$ images from 2 classes, dogs and cats, which is split into a $12,500$-image training set and a $12,500$-image test set. The **CIFAR-10** (Krizhevsky et al., 2014) dataset consists of $60,000$ $32 \times 32$ color images from 10 classes, which is split into a $50,000$-image training set and a $10,000$-image test set. The **LUNA16** (Setio et al., 2016) dataset consists of $888$ CT scans for lung nodule. We preprocessed the CT scans by generating $8106$ $50 \times 50$ gray-scale images, which is split into a $6484$-image training set and a $1622$-image testing set. **LUNA16** is highly imbalanced dataset ($85\%$, $15\%$).

**Synthesized crowdsourced labels in various of settings**    For each information structure in Figure 1, we generate two groups of crowdsourced labels for each dataset: labels provided by (H) experts with relatively high expertise; (L) experts with relatively low expertise. For each of the situation (H) (L), all three cases have the same senior experts.

**Case 4.1.** *(Independent mistakes) $M_s$ senior experts are mutually conditionally independent.*

**Case 4.2.** *(Naive majority) $M_s$ senior experts are mutually conditional independent, while other $M_j$ junior experts label all datapoints as the first class effortlessly.*

**Case 4.3.** *(Correlated mistakes) $M_s$ senior experts are mutually conditional independent, and each junior expert copies one of the senior experts.*

**Real-world dataset**    The **LabelMe** data (Rodrigues & Pereira, 2017; Russell et al., 2008) consists of a total of 2688 images, where 1000 of them were used to obtain labels from multiple annotators from Amazon Mechanical Turk and the remaining 1688 images were using for evaluating the different approaches. Each image was labeled by an average of 2.547 workers, with a mean accuracy of 69.2%.

**Networks**    We follow the four layers network in Rodrigues & Pereira (2017) on Dogs vs. Cats and LUNA16 and use VGG-16 on CIFAR-10 for the backbone of the data classifier $h$. For Labelme data, we apply the same setting of Rodrigues & Pereira (2017): we use pre-trained VGG-16 deep neural network and apply only one FC layer (with 128 units and ReLU activations) and one output layer on top, using 50% dropout.

We defer other implementation details to appendix B.

Table 2: Accuracy on LabelMe (real-world crowdsourced labels)

| Method | Majority Vote | Crowd Layer | Doctor Net | AggNet | Max-MIG |
|---|---|---|---|---|---|
| Accuracy | $80.41 \pm 0.56$ | $83.65 \pm 0.50$ | $80.56 \pm 0.59$ | $85.20 \pm 0.26$ | $\mathbf{86.42 \pm 0.36}$ |

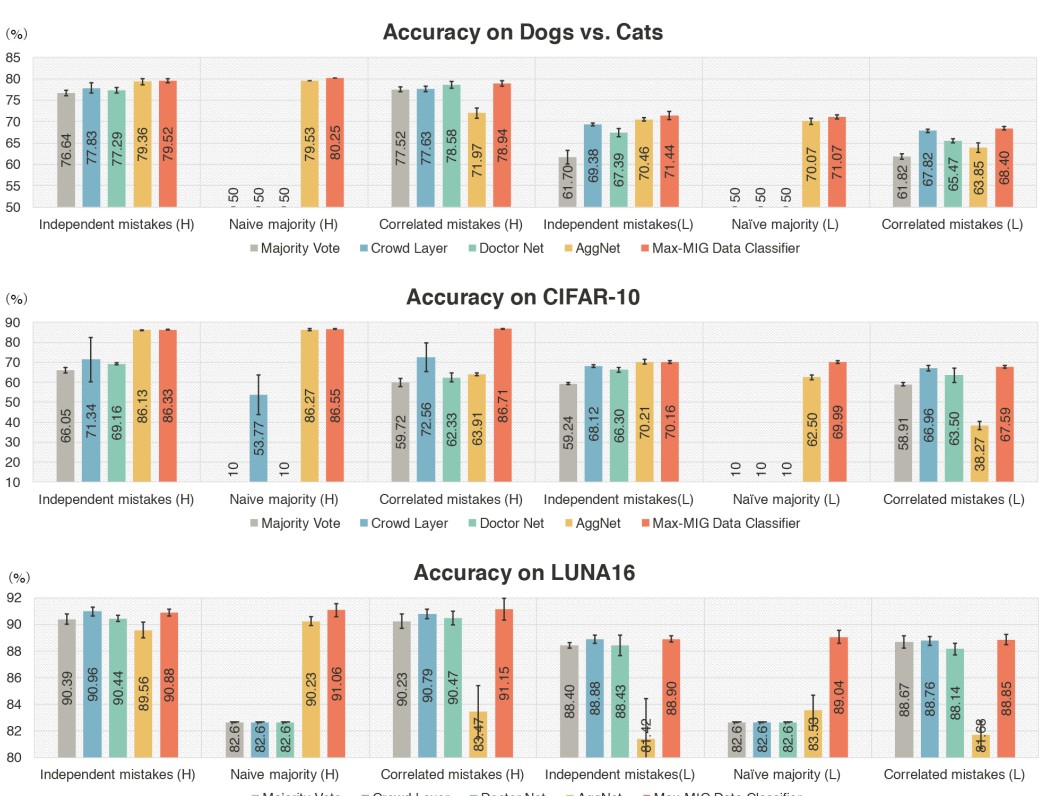

Figure 4: Results on Dogs vs. Cats, CIFAR-10, LUNA16.

## 4.1 RESULTS

We train the data classifier $h$ on the four datasets through our method[1] and other related methods. The accuracy of the trained data classifiers on the test set are shown in Table 2 and Figure 4. We also show the accuracy of our data-crowd forecaster and on the test set and compare it with AggNet (Table 3).

For the performances of the trained data classifiers, our method Max-MIG (red) almost outperform all other methods in every experiment. For the real-world dataset, LabelMe, we achieve the new state-of-the-art results. For the synthesized crowdsourced labels, the majority vote method (grey) fails in the naive majority situation. The AggNet has reasonably good performances when the experts are conditionally independent, including the naive majority case since naive expert is independent with everything, while it is outperformed by us a lot in the correlated mistakes case. This matches the theory in Appendix E: the AggNet is based on MLE and MLE fails in correlated mistakes case. The Doctor Net (green) and the Crowd Layer (blue) methods are not robust to the naive majority case. Our data-crowds forecaster (Table 3) performs better than our data classifier, which shows that our data-crowds forecaster actually takes advantage of the additional information, the crowdsourced labels, to give a better result. Like us, Aggnet also jointly trains the classifier and the aggregator, and can be used to train a data-crowds forecaster. We compared our data-crowds forecaster with Aggnet.

---

[1]The results of Max-MIG are based on KL divergence. The results for other divergences are similar.

The results still match our theory. When there is no correlated mistakes, we outperform Aggnet or have very similar performances with it. When there are correlated mistakes, we outperform Aggnet a lot (e.g. +30%).

Recall that in the experiments, for each of the situation (H) (L), all three cases have the same senior experts. Thus, all three cases' crowdsourced labels have the same amount of information. The results show that Max-MIG has similar performances for all three cases for each of the situation (H) (L), which validates our theoretical result: Max-MIG finds the "information intersection" between the data and the crowdsourced labels.

## 5 CONCLUSION AND DISCUSSION

We propose an information theoretic approach, Max-MIG, for joint learning from crowds, with a common assumption: the crowdsourced labels and the data are independent conditioning on the ground truth. We provide theoretical validation to our approach and compare our approach experimentally with previous methods (Doctor net (Guan et al., 2017), Crowd layer (Rodrigues & Pereira, 2017), Aggnet (Albarqouni et al., 2016)) under several different information structures. Each of the previous methods is not robust to at least one information structure and our method is robust to all and almost outperform all other methods in every experiment. To the best of our knowledge, our approach is the first algorithm that is both theoretically and empirically robust to the situation where some people make highly correlated mistakes and some people label effortlessly, without knowing the information structure among the crowds. We also test our method on real-world data and achieve the new state-of-the-art result.

Our current implementation of Max-MIG has several limitations. For example, we implement the aggregator using a simple linear model, which cannot handle the case when the senior experts are latent and cannot be linearly inferred from the junior experts. However, note that if the aggregator space is sufficiently rich, the Max-MIG approach is still able to handle any situation as long as the "information intersection" assumption holds. One potential future direction is designing more complicated but still trainable aggregator space.

ACKNOWLEDGMENTS

We would like to express our thanks for support from the following research grants NSFC-61625201 and 61527804.

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

## A  DATA-CROWDS FORECASTER COMPARISON

Table 3: Data-Crowds Forecaster Comparison: Max-MIG VS AggNet

| Dataset | Method | 4.1(H) | 4.2(H) | 4.3(H) | 4.1(L) | 4.2(L) | 4.3(L) |
|---|---|---|---|---|---|---|---|
| Dogs vs.Cats | Max-MIG (d) | 79.52 | 80.25 | 78.94 | 71.44 | 71.07 | 68.40 |
| | Max-MIG (dc) | 88.80 | 87.60 | 87.17 | 73.99 | 73.38 | 70.75 |
| | AggNet (d) | 79.36 | 79.53 | 71.97 | 70.46 | 70.07 | 63.85 |
| | AggNet (dc) | 88.00 | 88.56 | 75.00 | 71.27 | 70.75 | 61.14 |
| CIFAR-10 | Max-MIG(d) | 86.33 | 86.55 | 86.71 | 70.16 | 69.99 | 67.59 |
| | Max-MIG(dc) | 98.10 | 98.18 | 99.06 | 75.55 | 75.11 | 72.47 |
| | AggNet(d) | 86.13 | 86.27 | 63.91 | 70.21 | 62.50 | 38.27 |
| | AggNet(dc) | 99.05 | 99.01 | 70.01 | 74.76 | 72.02 | 29.03 |
| LUNA16 | Max-MIG(d) | 90.88 | 91.06 | 91.15 | 88.90 | 89.04 | 88.85 |
| | Max-MIG(dc) | 94.56 | 93.97 | 92.63 | 91.16 | 91.23 | 92.05 |
| | AggNet(d) | 89.56 | 90.23 | 83.47 | 81.42 | 83.53 | 81.68 |
| | AggNet(dc) | 91.13 | 91.94 | 65.14 | 70.97 | 74.41 | 61.76 |

Here (dc) is the shorthand for data-crowds forecaster and (d) is the shorthand for data-classifier. We take the average of five times experiments and the variance is pretty small. Due to space limitation, we omit the variance here.

## B    EXPERIMENTS DETAILS

### B.1    EXPERTS' EXPERTISE

For each information structure in Figure 1, we generate two groups of crowdsourced labels for each dataset: labels provided by (H) experts with relatively high expertise; (L) experts with relatively low expertise. For each of the situation (H) (L), all three cases have the same senior experts.

**Case B.1.** *(Independent mistakes) $M_s$ senior experts are mutually conditionally independent. (H) $M_s = 5$. (L) $M_s = 10$.*

**Dogs vs. Cats**    In situation (H), some senior experts are more familiar with cats, while others make better judgments on dogs. For example, expert A is more familiar with cats, her expertise for dogs/cats is 0.6/0.8 in the sense that if the ground truth is dog/cat, she labels the image as "dog"/"cat" with probability 0.6/0.8 respectively. Similarly, other experts expertise are B:0.6/0.6, C:0.9/0.6, D:0.7/0.7, E:0.6/0.7.

In situation (L), all ten seniors' expertise are 0.55/0.55.

**CIFAR-10**    In situation (H), we generate experts who may make mistakes in distinguishing the hard pairs: cat/dog, deer/horse, airplane/bird, automobile/trunk, frog/ship, but can perfectly distinguish other easy pairs (e.g. cat/frog), which makes sense in practice. When they cannot distinguish the pair, some of them may label the pair randomly and some of them label the pair the same class. In detail, for each hard pair, expert A label the pair the same class (e.g. A always labels the image as "cat" when the image has cats or dogs), expert B labels the pair uniformly at random (e.g. B labels the image as "cat" with the probability 0.5 and "dog" with the probability 0.5 when the image has cats or dogs). Expert C is familiar with mammals so she can distinguish cat/dog and deer/hose, while for other hard pairs, she label each of them uniformly at random. Expert D is familiar with vehicles so she can distinguish airplane/bird, automobile/trunk and frog/ship, while for other hard pairs, she always label each of them the same class. Expert E does not have special expertise. For each hard pair, Expert E labels them correctly with the probability 0.6.

In situation (L), all ten senior experts label each image correctly with probability 0.2 and label each image as other false classes uniformly with probability $\frac{0.8}{9}$.

**LUNA16**    In situation (H), some senior experts tend to label the image as "benign" while others tend to label the image as "malignant". Their expertise for benign/malignant are: A: 0.6/0.9, B:0.7/0.7, C:0.9/0.6, D:0.6/0.7, E:0.7/0.6.

In situation (L), all ten seniors' expertise are 0.6/0.6.

**Case B.2.** *(Naive majority) $M_s$ senior experts are mutually conditional independent, while other $M_j$ junior experts label all data as the first class effortlessly. (H) $M_s = 5$, $M_j = 5$. (L) $M_s = 10$, $M_j = 15$.*

For Dogs vs. Cats, all junior experts label everything as "cat". For CIFAR-10, all junior experts label everything as "airplane". For LUNA16, all junior experts label everything as "benign".

**Case B.3.** *(Correlated mistakes) $M_s$ senior experts are mutually conditional independent, and each junior expert copies one of the senior experts.(H) $M_s = 5$, $M_j = 5$. (L) $M_s = 10$, $M_j = 2$.*

For Dogs vs. Cats, CIFAR-10 and LUNA16, in situation (H), two junior experts copy expert $A$'s labels and three junior experts copy expert $C$'s labels; in situation (L), one junior expert copies expert $A$'s labels and another junior expert copies expert $C$'s labels.

### B.2    IMPLEMENTATION DETAILS

**Networks**    For Dogs vs. Cats and LUNA16, we follow the four layers network in Rodrigues & Pereira (2017). We use Adam optimizer with learning rate $1.0 \times 10^{-4}$ for both the data classifier and the crowds aggregator. Batch size is set to 16. For CIFAR-10, we use VGG-16 as the backbone. We use Adam optimizer with learning rate $1.0 \times 10^{-3}$ for the data classifier and $1.0 \times 10^{-4}$ for the crowds aggregator. Batch size is set to 64.

For Labelme data, We apply the same setting of Rodrigues & Pereira (2017): we use pre-trained VGG-16 deep neural network and apply only one FC layer (with 128 units and ReLU activations) and one output layer on top, using 50% dropout. We use Adam optimizer with learning rate $1.0 \times 10^{-4}$ for both the data classifier and the crowds aggregator.

For our method MAX-MIG's crowds aggregator, for Dogs vs. Cats and LUNA16, we set the bias $\mathbf{b}$ as $\log \mathbf{p}$ and only tune $\mathbf{p}$. For CIFAR-10 and Labelme data, we fix the prior distribution $\mathbf{p}$ to be the uniform distribution $\mathbf{p}_0$ and fix the bias $\mathbf{b}$ as $\log \mathbf{p}_0$.

**Initialization**    For AggNet and our method Max-MIG, we initialize the parameters $\{\mathbf{W}_m\}_m$ using the method in Raykar et al. (2010):

$$W_{c,c'}^m = \log \frac{\sum\limits_{i=1}^{N} Q(y_i = c)\mathbb{1}(y_i^m = c')}{\sum\limits_{i=1}^{N} Q(y_i = c)} \tag{2}$$

where $\mathbb{1}(y_i^m = c') = 1$ when $y_i^m = c'$ and $\mathbb{1}(y_i^m = c') = 0$ when $y_i^m \neq c'$ and N is the total number of datapoints. We average all crowdsourced labels to obtain $Q(y_i = c) := \frac{1}{M} \sum\limits_{m=1}^{M} \mathbb{1}(y_i^m = c)$.

For Crowd Layer method, we initialize the weight matrices using identity matrix on Dogs vs. Cats and LUNA as Rodrigues & Pereira (2017) suggest. However, this initialization method leads to pretty bad results on CIFAR-10. Thus, we use (2) for Crowd Layer on CIFAR-10, which is the best practice in our experiments.

## C $f$-MUTUAL INFORMATION

### C.1 $f$-DIVERGENCE AND FENCHEL'S DUALITY

$f$-**divergence (Ali & Silvey, 1966; Csiszár et al., 2004)**    $f$-divergence $D_f : \Delta_\Sigma \times \Delta_\Sigma \mapsto \mathbb{R}$ is a non-symmetric measure of the difference between distribution $\mathbf{p} \in \Delta_\Sigma$ and distribution $\mathbf{q} \in \Delta_\Sigma$ and is defined to be

$$D_f(\mathbf{p}, \mathbf{q}) = \sum_{\sigma \in \Sigma} \mathbf{p}(\sigma) f\left(\frac{\mathbf{q}(\sigma)}{\mathbf{p}(\sigma)}\right)$$

where $f : \mathbb{R} \mapsto \mathbb{R}$ is a convex function and $f(1) = 0$.

### C.2 $f$-MUTUAL INFORMATION

Given two random variables $X, Y$ whose realization space are $\Sigma_X$ and $\Sigma_Y$, let $\mathbf{U}_{X,Y}$ and $\mathbf{V}_{X,Y}$ be two probability measures where $\mathbf{U}_{X,Y}$ is the joint distribution of $(X, Y)$ and $\mathbf{V}_{X,Y}$ is the product of the marginal distributions of $X$ and $Y$. Formally, for every pair of $(x, y) \in \Sigma_X \times \Sigma_Y$,

$$\mathbf{U}_{X,Y}(X = x, Y = y) = \Pr[X = x, Y = y] \qquad \mathbf{V}_{X,Y}(X = x, Y = y) = \Pr[X = x]\Pr[Y = y].$$

If $\mathbf{U}_{X,Y}$ is very different from $\mathbf{V}_{X,Y}$, the mutual information between $X$ and $Y$ should be high since knowing $X$ changes the belief for $Y$ a lot. If $\mathbf{U}_{X,Y}$ equals to $\mathbf{V}_{X,Y}$, the mutual information between $X$ and $Y$ should be zero since $X$ is independent with $Y$. Intuitively, the "distance" between $\mathbf{U}_{X,Y}$ and $\mathbf{V}_{X,Y}$ represents the mutual information between them.

**Definition C.1** ($f$-mutual information (Kong & Schoenebeck, 2016))**.** *The $f$-mutual information between $X$ and $Y$ is defined as*

$$MI^f(X, Y) = D_f(\mathbf{U}_{X,Y}, \mathbf{V}_{X,Y})$$

*where $D_f$ is $f$-divergence. $f$-mutual information is always non-negative.*

Kong & Schoenebeck (2016) show that if we measure the amount of information by $f$-mutual information, any "data processing" on either of the random variables will decrease the amount of information crossing them. With this property, Kong & Schoenebeck (2016) propose an information theoretic mechanism design framework using $f$-mutual information. Kong & Schoenebeck (2018) reduce the co-training problem to a mechanism design problem and extend the information theoretic framework in Kong & Schoenebeck (2016) to address the co-training problem.

## D  PROOF OF THEOREM 3.4

This section provides the formal proofs to our main theorem.

**Definition D.1** (Confusion matrix). *For each expert $m$, we define her confusion matrix as* $\mathbf{C}^m = (C_{c,c'}^m)_{c,c'} \in \mathbb{R}^{|\mathcal{C}| \times |\mathcal{C}|}$ *where* $C_{c,c'}^m = P(Y^m = c'|Y = c)$.

We denote the set of all possible classifiers by $H_\infty$ and the set of all possible aggregators by $G_\infty$.

**Lemma D.2.** *(Kong & Schoenebeck, 2018) With assumption 3.1, 3.3, $(h^*, g^*, \mathbf{p}^*)$ is a maximizer of*

$$\max_{h \in H_\infty, g \in G_\infty, \mathbf{p} \in \Delta_\mathcal{C}} \mathbb{E}_{X, Y^{[M]}} MIG^f(h(X), g(Y^{[M]}), \mathbf{p})$$

*and the maximum is the $f$ mutual information between $X$ and $Y^{[M]}$, $MI^f(X, Y^{[M]})$. Moreover, $\zeta^*(x, y^{[M]}) = \zeta(x, y^{[M]}; h^*, g^*, \mathbf{p}^*)$ for every $x, y^{[M]}$.*

**Proposition D.3.** *[Independent mistakes] With assumptions 3.1, 3.3, if the experts are mutually independent conditioning on $Y$, then $g^* \in G_{WA}$ and*

$$g^*(y^{[M]}) = g(y^{[M]}; \{\log \mathbf{C}^m\}_{m=1}^M, \log \mathbf{p}^*)$$

*for every $y^{[M]} \in \mathcal{C}^M$.*

*This implies that $(h^*, g^*, \mathbf{p}^*)$ is a maximizer of*

$$\max_{h \in H_{NN}, g \in G_{WA}, \mathbf{p} \in \Delta_\mathcal{C}} \mathbb{E}_{X, Y^{[M]}} MIG^f(h(X), g(Y^{[M]}), \mathbf{p})$$

*and the maximum is the $f$ mutual information between $X$ and $Y^{[M]}$, $MI^f(X, Y^{[M]})$. Moreover, $\zeta^*(x, y^{[M]}) = \zeta(x, y^{[M]}; h^*, g^*, \mathbf{p}^*)$ for every $x, y^{[M]}$.*

*Proof.* We will show that when the experts are mutually conditionally independent, then

$$g^*(y^{[M]}) = g(y^{[M]}; \{\log \mathbf{C}^m\}_{m=1}^M, \log \mathbf{p}^*).$$

This also implies that $g^* \in G_{WA}$. Based on the result of Lemma D.2, by assuming that $h^* \in H_{NN}$, we can see $(h^*, g^*, \mathbf{p}^*)$ is a maximizer of $\max_{h \in H_{NN}, g \in G_{WA}, \mathbf{p} \in \Delta_\mathcal{C}} MIG^f(h, g, \mathbf{p})$ and the maximum is the $f$ mutual information between $X$ and $Y^{[M]}$. Moreover, Lemma D.2 also implies that $\zeta^*(x, y^{[M]}) = \zeta(x, y^{[M]}; h^*, g^*, \mathbf{p}^*)$ for every $x, y^{[M]}$.

For every $c \in \mathcal{C}$, every $y^{[M]} \in \mathcal{C}^M$,

$$(\log g^*(y^{[M]}))_c = \log P(Y = c|Y^{[M]} = y^{[M]})$$
$$= \log P(Y^{[M]} = y^{[M]}|Y = c)P(Y = c) - \log P(Y^{[M]} = y^{[M]})$$
$$= \sum_{m=1}^M \log P(Y^m = y^m|Y = c) + \log P(Y = c) - \log P(Y^{[M]} = y^{[M]})$$

Thus,

$$(\sum_{m=1}^M \log \mathbf{C}^m \cdot \mathbf{e}^{(y^m)} + \log \mathbf{p}^*)_c = \sum_{m=1}^M \log P(Y^m = y^m|Y = c) + \log P(Y = c)$$
$$= (\log g^*(y^{[M]}))_c + \log P(Y^{[M]} = y^{[M]})$$

Then,

$$
\begin{aligned}
(softmax(\sum_m \log \mathbf{C}^m \cdot \mathbf{e}^{(y^m)} + \log \mathbf{p}^*))_c &= \frac{e^{(\log g^*(y^{[M]}))_c + \log P(Y^{[M]} = y^{[M]})}}{\sum_c e^{(\log g^*(y^{[M]}))_c + \log P(Y^{[M]} = y^{[M]})}} \\
&= \frac{e^{(\log g^*(y^{[M]}))_c}}{\sum_c e^{(\log g^*(y^{[M]}))_c}} \\
&= (g^*(y^{[M]}))_c \\
&\quad (\text{since } g^*(y^{[M]}) \in \Delta_{\mathcal{C}}, \sum_c g^*(y^{[M]})_c = 1)
\end{aligned}
$$

Thus,

$$
g^*(y^{[M]}) = softmax(\sum_m \log \mathbf{C}^m \cdot \mathbf{e}^{(y^m)} + \log \mathbf{p}^*) = g(y^{[M]}; \{\log \mathbf{C}^m\}_{m=1}^M, \log \mathbf{p}^*).
$$

$\square$

We restate our main theorem, Theorem 3.4, here with more details and prove it.

**Theorem 3.4** (General case). *With assumption 3.1, 3.3, when there exists a subset of experts $\mathcal{S} \subset [M]$ such that the experts in $\mathcal{S}$ are mutually independent conditioning on $Y$ and $Y^{\mathcal{S}}$ is a sufficient statistic for $Y$, i.e. $P(Y = y|Y^{[M]} = y^{[M]}) = P(Y = y|Y^{\mathcal{S}} = y^{\mathcal{S}})$ for every $y \in \mathcal{C}, y^{[M]} \in \mathcal{C}^M$, then $g^* \in G_{WA}$ and*

$$
g^*(y^{[M]}) = g(y^{[M]}; \{\mathbf{W}^{*m}\}_m, \log \mathbf{p}^*)
$$

*for every $y^{[M]} \in \mathcal{C}^M$ where for every $m \in \mathcal{S}$, $\mathbf{W}^{*m} = \log \mathbf{C}^m$, for every $m \notin \mathcal{S}$, $\mathbf{W}^{*m} = \mathbf{0}^2$.*

*This implies that $(h^*, g^*, \mathbf{p}^*)$ is a maximizer of*

$$
\max_{h \in H_{NN}, g \in G_{WA}, \mathbf{p} \in \Delta_{\mathcal{C}}} \mathbb{E}_{X, Y^{[M]}} MIG^f(h(X), g(Y^{[M]}), \mathbf{p})
$$

*and the maximum is the $f$ mutual information between $X$ and $Y^{[M]}$, $MI^f(X, Y^{[M]})$. Moreover, $\zeta^*(x, y^{[M]}) = \zeta(x, y^{[M]}; h^*, g^*, \mathbf{p}^*)$ for every $x, y^{[M]}$.*

*Proof.* Like the proof for the above proposition, we need to show that

$$
g^*(y^{[M]}) = g(y^{[M]}; \{\mathbf{W}^{*m}\}_m, \log \mathbf{p}^*).
$$

This also implies that $g^* \in G_{WA}$ as well as the other results of the theorem.

When $Y^{\mathcal{S}}$ is a sufficient statistic for $Y$, we have

$$
g^*(y^{[M]}) = g^*(y^{\mathcal{S}}).
$$

Proposition D.3 shows that

$$
g^*(y^{\mathcal{S}}) = g(y^{\mathcal{S}}; \{\log \mathbf{C}^s\}_{s \in \mathcal{S}}, \log \mathbf{p}^*).
$$

Thus, we have

$$
g^*(y^{[M]}) = g^*(y^{\mathcal{S}}) = g(y^{\mathcal{S}}; \{\log \mathbf{C}^s\}_{s \in \mathcal{S}}, \log \mathbf{p}^*) = g(y^{[M]}; \{\mathbf{W}^{*m}\}_m, \log \mathbf{p}^*)
$$

where for every $m \in \mathcal{S}$, $\mathbf{W}^{*m} = \log \mathbf{C}^m$, for every $m \notin \mathcal{S}$, $\mathbf{W}^{*m} = \mathbf{0}$.

$\square$

---

[2]We denote the matrix whose entries are all zero by $\mathbf{0}$.

# E  THEORETICAL COMPARISONS WITH MLE

Raykar et al. (2010) propose a maximum likelihood estimation (MLE) based method in the learning from crowds scenario. Raykar et al. (2010) use logistic regression and Aggnet(Albarqouni et al., 2016) extends it to combine with the deep learning model. In this section, we will theoretically show that these MLE based methods can handle the independent mistakes case but cannot handle even the simplest correlated mistakes case—only one expert reports meaningful information and all other experts always report the same meaningless information—which can be handled by our method. Therefore, in addition to the experimental results, theoretically, our method is still better than these MLE based methods. We first introduce these MLE based methods.

Let $\Theta$ be the parameter that control the distribution over $X$ and $Y$. Let $\Theta^m$ be the parameter that controls the distribution over $Y^m$ and $Y$.

For each each $x$, $y^{[M]}$,

$$
\begin{aligned}
&P(Y^{[M]} = y^{[M]} | X = x; \Theta, \{\Theta^m\}_m) \quad\quad\quad (3)\\
&= \sum_y P(Y = y | X = x; \Theta) P(Y^{[M]} = y^{[M]} | Y = y; \{\Theta^m\}_m)
\end{aligned}
$$

(conditioning on $Y$, $X$ and $Y^{[M]}$ are independent)

$$
= \sum_y P(Y = y | X = x; \Theta) \Pi_{m=1}^M P(Y^m = y^m | Y = y; \Theta^m)
$$

(experts are mutually conditional independent.)

The MLE based method seeks $\Theta$ and $\Theta^m$ that maximize

$$
\sum_{i=1}^N \log \sum_c P(Y = c | X = x_i; \Theta) \Pi_{m=1}^M P(Y_i^m = y_i^m | Y = c; \Theta^m)
$$

To theoretically compare it with our method, we use our language to reinterpret the above MLE based method.

We define $\mathcal{T}$ as the set of all $|\mathcal{C}| \times |\mathcal{C}|$ transition matrices with each row summing to 1.

For each expert $m$, we define $\mathbf{W}^m \in \mathcal{T}$ as a parameter that is associated with $m$.

Given a set of data classifiers $h \in H$ where $h : I \mapsto \Delta_\mathcal{C}$, the MLE based method seeks $h \in H$ and transition matrices $\mathbf{W}^1, \mathbf{W}^2, \cdots, \mathbf{W}^M \in \mathcal{T}$ that maximize

$$
\sum_{i=1}^N \log \sum_c h(x_i)_c \Pi_{m=1}^M W_{c,y_i^m}^m.
$$

The expectation of the above formula is

$$
\mathbb{E}_{X,Y^{[M]}} \log \sum_c h(X)_c \Pi_{m=1}^M W_{c,Y^m}^m.
$$

Note that Raykar et al. (2010) set the data classifiers space $H$ as all logistic regression classifiers and Albarqouni et al. (2016) extend this space to the neural network space.

**Proposition E.1** (MLE works for independent mistakes). *If the experts are mutually independent conditioning on Y, then $h^*$ and $\mathbf{C}^1, \mathbf{C}^2, \cdots, \mathbf{C}^M$ are a maximizer of*

$$
\max_{h,\mathbf{W}^1,\mathbf{W}^2,\cdots,\mathbf{W}^m \in \mathcal{T}} \mathbb{E}_{X,Y^{[M]}} \log \sum_c h(X)_c \Pi_{m=1}^M W_{c,Y^m}^m.
$$

*Proof.*

$$\mathbb{E}_{X,Y^{[M]}} \log \sum_c h(X)_c \Pi_{m=1}^M W_{c,Y^m}^m$$

$$= \sum_{x,y^{[M]}} P(X = x, Y^{[M]} = y^{[M]}) \log \sum_c h(x)_c \Pi_{m=1}^M W_{c,y^m}^m$$

$$= \sum_x P(X = x) \sum_{y^{[M]}} P(Y^{[M]} = y^{[M]} | X = x) \log \sum_c h(x)_c \Pi_{m=1}^M W_{c,y^m}^m$$

Since $\mathbf{W}^1, \mathbf{W}^2, \cdots, \mathbf{W}^m \in \mathcal{T}$, thus,

$$\sum_{y^{[M]} \in \mathcal{C}^M} \sum_{c \in \mathcal{C}} h(x)_c \Pi_{m=1}^M W_{c,y^m}^m = 1$$

which means $(\sum_{c \in \mathcal{C}} h(x)_c \Pi_{m=1}^M W_{c,y^m}^m)_{y^{[M]}}$ can be seen as a distribution over all possible $y^{[M]} \in \mathcal{C}^M$. Moreover, for any two distribution vectors $\mathbf{p}$ and $\mathbf{q}$, $\mathbf{p} \cdot \log \mathbf{q} \le \mathbf{p} \cdot \log \mathbf{p}$, thus

$$\sum_x P(X = x) \sum_{y^{[M]}} P(Y^{[M]} = y^{[M]} | X = x) \log \sum_c h(x)_c \Pi_{m=1}^M W_{c,y^m}^m$$

$$\le \sum_x P(X = x) \sum_{y^{[M]}} P(Y^{[M]} = y^{[M]} | X = x) \log P(Y^{[M]} = y^{[M]} | X = x)$$

$$= \sum_x P(X = x) \sum_{y^{[M]}} P(Y^{[M]} = y^{[M]} | X = x) \log \sum_c h^*(x)_c \Pi_{m=1}^M C_{c,Y^m}^m \qquad \text{(see equation (3))}$$

$\square$

Thus, the MLE based method handles the independent mistakes case. However, we will construct a counter example to show that it cannot handle a simple correlated mistakes case which can be handled by our method.

**Example E.2** (A simple correlated mistakes case). *We assume there are only two classes $\mathcal{C} = \{0, 1\}$ and the prior over $Y$ is uniform, that is, $P(Y = 0) = P(Y = 1) = 0.5$. We also assume that $X = Y$.*

*There are 101 experts and one of the experts, say her the first expert, fully knows $Y$ and always reports $Y^1 = Y$. The second expert knows nothing and every time flips a random unbiased coin whose randomness is independent with $X, Y$. She reports $Y^2 = 1$ when she gets head and reports $Y^2 = 0$ otherwise. The rest of experts copy the second expert's answer all the time, i.e. $Y^m = Y^2$, for $m \ge 2$.*

Note that our method can handle this simple correlated mistakes case and will give all useless experts weight zero based on Theorem 3.4.

We define $h_0$ as a data classifier such that $h_0(x)_0 = h_0(x)_1 = 0.5$. We will show this meaningless data classifier $h_0$ has much higher likelihood than $h^*$, which shows that in this simple correlated mistakes case, the MLE based method will obtain meaningless results.

We define a data classifier $h$'s maximal expected likelihood as

$$\max_{\mathbf{W}^1, \mathbf{W}^2, \cdots, \mathbf{W}^m \in \mathcal{T}} \mathbb{E}_{X,Y^{[M]}} \log \sum_c h(X)_c \Pi_{m=1}^M W_{c,Y^m}^m.$$

**Theorem E.3** (MLE fails for correlated mistakes). *In the scenario defined by Example E.2, the meaningless classifier $h_0$'s maximal expected likelihood is at least $\log 0.5$ and the Bayesian posterior classifier $h^*$'s maximal expected likelihood is $100 \log 0.5 \ll \log 0.5$.*

The above theorem implies that the MLE based method fails in Example E.2.

*Proof.* For the Bayesian posterior classifier $h^*$, since $X = Y = Y^1$ and $Y^2 = \cdots = Y^M$, then $h^*(X = c)$ is an one-hot vector where the $c^{th}$ entry is 1 and everything is determined by the realizations of $Y$ and $Y^2$.

$$\mathbb{E}_{X,Y^{[M]}} \log \sum_c h^*(X)_c \Pi_{m=1}^M W_{c,y^m}^m$$

$$= \sum_{x,y^{[M]}} P(X = x, Y^{[M]} = y^{[M]}) \log \sum_c h^*(x)_c \Pi_{m=1}^M W_{c,y^m}^m$$

$$= \sum_{c,y^{[M]}} P(X = c, Y^{[M]} = y^{[M]}) \log \sum_c h^*(c)_c \Pi_{m=1}^M W_{c,y^m}^m$$

$$= \sum_{c,c'} P(Y = c) P(Y^2 = c') \log W_{c,c}^1 \Pi_{m=2}^M W_{c,c'}^m \qquad (X = Y = Y^1, Y^2 = \cdots = Y^M)$$

$$= \sum_c P(Y = c) \log W_{c,c}^1 + \sum_{m=2}^M \sum_c P(Y = c) \sum_{c'} P(Y^2 = c') \log W_{c,c'}^m$$

$$\leq \sum_{m=2}^M \sum_c P(Y = c) \sum_{c'} P(Y^2 = c') \log W_{c,c'}^m$$

$$\leq \sum_{m=2}^M \sum_c P(Y = c) \sum_{c'} P(Y^2 = c') \log P(Y^2 = c')$$

$$\qquad\qquad (W^m \text{ is a transition matrix and } \mathbf{p} \cdot \log \mathbf{q} \leq \mathbf{p} \cdot \log \mathbf{p})$$

$$= 100 \log 0.5 \qquad (Y^2 \text{ equals 0 with probability 0.5 and 1 with probability 0.5 as well})$$

The maximal value is obtained by setting $\mathbf{W}^1$ as an identity matrix and setting $\mathbf{W}^2 = \cdots = \mathbf{W}^M$ as $\begin{pmatrix} 0.5 & 0.5 \\ 0.5 & 0.5 \end{pmatrix}$. Thus, the Bayesian posterior data classifier $h^*$'s maximal expected likelihood is $100 \log 0.5$. For the meaningless data classifier $h_0$,

$$\mathbb{E}_{X,Y^{[M]}} \log \sum_c h_0(X)_c \Pi_{m=1}^M W_{c,y^m}^m$$

$$= \sum_{x,y^{[M]}} P(X = x, Y^{[M]} = y^{[M]}) \log \sum_c h_0(x)_c \Pi_{m=1}^M W_{c,y^m}^m$$

$$= \sum_{x,y^{[M]}} P(X = x, Y^{[M]} = y^{[M]}) \log 0.5 \sum_c \Pi_{m=1}^M W_{c,y^m}^m$$

$$= \sum_{c,c'} P(Y = c) P(Y^2 = c') \log 0.5 \sum_c \Pi_{m=1}^M W_{c,c'}^m$$

Note when we set every $\mathbf{W}^m$ as an identity matrix, the above formula equals $\log 0.5$. Thus, the meaningless data classifier $h_0$'s maximal expected likelihood is at least $\log 0.5$.

$\square$

