# OpenReview forum: "Max-MIG: an Information Theoretic Approach for Joint Learning from Crowds"
_ICLR.cc/2019/Conference_

### Official Review · AnonReviewer1 · 2018-10-31
**Updated review**

**Rating:** 7
**Confidence:** 4

**Review:**

EDIT: I thank the authors for providing all clarifications. I think this paper is a useful contribution. It will be of interest to the audience in the conference.

Summary:
This paper provides a method to jointly learn from crowdsourced worker labels and the actual data. The key claimed difference is that previous works on crowdsourced worker labels ignored the data. At a higher level, the algorithm comprises maximizing the mutual information gain between the worker labels and the output of a neural network (or more generally any ML model) on the data.

Evaluation:
I like the idea behind the algorithm. However there are several issues on which I ask the authors to provide some clarity. I will provide a formal "evaluation" after that. (For the moment, please ignore the "rating". I will provide one after the rebuttal.)

(1) As the authors clarified, one key aspect of the "information intersection" assumption is that the crowdsourced labels are statistically independent from the data when conditioned on the ground truth. How strongly does this coincide with reality? Since the work is primary empirical, is there any evidence on this front?

(2) In the abstract, introduction etc., what does it mean to say that the algorithm is an "early algorithm"?
-- Thanks for the clarification. I would suggest using the term "first algorithm" in such cases. However, is this the first algorithm towards this goal? See point (3).

(3) The submitted paper misses an extremely relevant piece of literature: "Learning From Noisy Singly-labeled Data" (arXiv:1712.04577). This paper also aims to solve the label + features problem together. How do the results of this paper compare to that of this submission?

(4) "Model and assumptions" Is the i.i.d. assumption across the values of "i"? Then does that not violate the earlier claim of accommodating correlated mistakes?

(5) Recent papers on crowdsourcing (such as Achieving budget-optimality with adaptive schemes in crowdsourcing arXiv:1602.03481 and  A Permutation-based Model for Crowd Labeling: Optimal Estimation and Robustness arXiv:1606.09632) go beyond restricting workers to have a common confusion matrix for all questions. In this respect, these are better aligned with the realistic scenario where the error in labeling may depend on the closeness to the decision boundary. How do these settings and algorithms relate to the submission?

(6) Page 5: "Later we will show...."   Later where? Please provide a reference.

(7) Theorem 3.4, The assumption of existence of experts such that Y^S is a sufficient statistic for Y: For instance, suppose there are 10 experts who all have a 0.999 probability of correctness (assume symmetric confusion matrices) and there are 5 non-experts who have a 0.001 probability of correctness and even if we suppose all are mutually independent given the true label, then does this satisfy this sufficient statistic assumption? This appears to be a very strong assumption, but perhaps the authors have better intuition?

(8) The experiments comprise only some simulations. The main point of experiments (particularly in the absence of any theoretical results) towards bolstering the paper is to ensure that the assumptions are at least somewhat reasonable. I believe there are several datasets collected from Amazon Mechanical Turk available online? Otherwise, would it be possible to run realistic experiments on some crowdsourcing platforms?

---

> ### Author Response · Authors · 2018-11-13
> **Thanks for your review and here are our clarifications.**
>
> Thank you for your review and comments.
>
> Q: >>As the authors clarified, one key aspect of the "information intersection" assumption is that the crowdsourced labels are statistically independent from the data when conditioned on the ground truth. How strongly does this coincide with reality? Since the work is primary empirical, is there any evidence on this front?
>
> A: <<1) Let's consider the case where we ask the turkers to label "dogs vs cats". This assumption says that the turkers' labels are the noisy version of the ground truth class and the noise is independent with other aspects of the images (e.g. the image scene is indoor or outdoor). When the assumption is violated in the sense that the turkers' noises are highly correlated with other aspects of the images (e.g. the image scene is indoor or outdoor), without other assumptions, no algorithm can train a classifier here to avoid the influence of the ``indoor or outdoor'' information.
> 2)This assumption is commonly used in most crowd-learning literature  ( Dawid & Skene (1979), Raykar et al. (2010),  Albarqouni et al. (2016),  Guan et al. (2017) , Rodrigues & Pereira (2017) ).
>
> Q: >>is this the first algorithm towards this goal?
> A: <<It's not the first algorithm to ``joint'' learn (Raykar et al. 2010 is the first). It is the first algorithm that is robust to various information structures theoretically and experimentally. Learning From Noisy Singly-labeled Data is not robust to correlated mistakes (see following detailed comparison).
>
> Q: >>The submitted paper misses an extremely relevant piece of literature: "Learning From Noisy Singly-labeled Data" (arXiv:1712.04577). This paper also aims to solve the label + features problem together. How do the results of this paper compare to that of this submission?
> A: <<Thanks for your information. We will cite this ICLR 18 paper. Theoretically, this paper still requires the experts to be mutually conditional independent while we do not. Empirically, we tested this method on LabelMe data which has the real Amazon MTurk crowd labels and our method still outperforms this method: Max-MIG 86.42 +/- 0.36,  MBEM(ICLR 18) 81.24 +/- 1.60.
>
> Q: >>"Model and assumptions" Is the i.i.d. assumption across the values of "i"? Then does that not violate the earlier claim of accommodating correlated mistakes?
> A: <<It means {（x_1,y_1^1,...,y_1^M),（x_2,y_2^1,...,y_2^M),...} =（x_i,y_i^1,...,y_i^M)_i=are i.i.d. samples of the joint random variables (X,Y^1,....,Y^M). A non-i.i.d example is that all（x_i,y_i^1,...,y_i^M)_i are the same. Experts make correlated mistakes means the random variables Y^1,...,Y^M are correlated even conditioning on the ground truth. There is no contradiction here.
>
> Q: >>Recent papers on crowdsourcing (such as Achieving budget-optimality with adaptive schemes in crowdsourcing arXiv:1602.03481 and  A Permutation-based Model for Crowd Labeling: Optimal Estimation and Robustness arXiv:1606.09632) go beyond restricting workers to have a common confusion matrix for all questions. In this respect, these are better aligned with the realistic scenario where the error in labeling may depend on the closeness to the decision boundary. How do these settings and algorithms relate to the submission?
> A: <<Thanks for your information. One possible direction is making both the ground truth and image difficulty as the information intersection and finds them. We agree that taking account of image difficulty is an interesting direction to explore in future work and we will try to combine our framework with relevant papers in the future.
>
> Q: >>Page 5: "Later we will show...."   Later where? Please provide a reference.
> A: <<The formal statement is in Appendix C, Theorem 3.4 (this is a detailed statement compared with the Theorem 3.4 in the main body). We will clarify it in our revised paper.
>
> Q: >>Theorem 3.4, The assumption of existence of experts such that Y^S is a sufficient statistic for Y: For instance, suppose there are 10 experts who all have a 0.999 probability of correctness (assume symmetric confusion matrices) and there are 5 non-experts who have a 0.001 probability of correctness and even if we suppose all are mutually independent given the true label, then does this satisfy this sufficient statistic assumption? This appears to be a very strong assumption, but perhaps the authors have better intuition?
> A: <<1) Our algorithm can handle all mutually independent cases, which includes the above .999 example since the mutually independent case satisfies our assumption automatically where all experts can be seen as senior experts (see Proposition C.3. for detail). We will clarify this in our revised paper. 2)We test our algorithm in real data (see our top comments) and the results show that our algorithm is robust to the real case.
>
>
> Q: >>Realistic experiments on some crowdsourcing platforms?
> A: <<See our top comments.

---

> > ### Comment · AnonReviewer1 · 2018-11-14
> > **Follow-up questions**
> >
> > Thank you for your response. The experimental results are indeed very positive. I have two follow-up comments:
> >
> > - Regarding earlier question (7): My question is about Theorem 3.4. I repeat my question: Does the example in my question satisfy the sufficient statistic condition? If so, then is there an easy way to see that it does? If not, then is the main theorem missing a very important case (and perhaps calls for the later proposition to be brought into the main text)? If not, also then what is the implication and/or meaning of this sufficient statistic condition? Unfortunately, this apparently important requirement is quite hidden within all notation etc. In the revision, please clarify the meaning and implications of this condition (in the main text or appendix).
> >
> > - Regarding earlier questions (1), (3), (5): The simple setting of (5), which is highly prevalent in practice, does NOT satisfy the "information intersection" assumption. I am fine with this assumption since it appears often in earlier works in crowdsourced labeling but the paper needs to be very clear in the benefits as well as the limitations of this assumption. In the revision, please make a very careful comparison of pros and cons with respect to the references in (3) and (5).

---

> > > ### Author Response · Authors · 2018-11-18
> > > **Reply**
> > >
> > > Thank you for your questions.
> > >
> > > Q: My question is about Theorem 3.4. I repeat my question: Does the example in my question satisfy the sufficient statistic condition? If so, then is there an easy way to see that it does?
> > > A: Yes. The short explanation is: in your example, we can make the set of senior experts S consist of both the experts and the non-experts.
> > >
> > > The long explanation is: in the example of your question, all experts are mutually independent conditioning on the ground truth. All independent mistakes cases satisfy SSC since (1) SSC requires that there EXISTs a subset of experts S (we call them senior experts), whose identities are unknown, such that the experts in S have mutually independent labeling biases and it is sufficient to only use the experts in S’ information to predict the ground truth label; (2) in the independent mistakes case, we can make S = M, where M is the set of all experts.
> > >
> > > Q: Unfortunately, this apparently important requirement is quite hidden within all notation etc. In the revision, please clarify the meaning and implications of this condition (in the main text or appendix).
> > > A: Thanks for your suggestion. In addition to the explanation of this requirement in the last paragraph of our intro, in our revised version, we have also clarified it after the statement of our main theorem.
> > >
> > > Q:  The simple setting of (5), which is highly prevalent in practice, does NOT satisfy the "information intersection" assumption. I am fine with this assumption since it appears often in earlier works in crowdsourced labeling but the paper needs to be very clear in the benefits as well as the limitations of this assumption. In the revision, please make a very careful comparison of pros and cons with respect to the references in (3) and (5).
> > >
> > > A: The "information intersection" assumption involves both the crowdsourced labels and the datapoints while both settings of (5),  Khetan & Oh 2016 and Shah et al. 2016, are for pure crowdsourcing methods. Thus, we assume that this question means the settings of (5), i.e. Khetan & Oh 2016 and Shah et al. 2016, do not use the original Dawid-Skene model to model the experts while our crowdsourcing part uses the original Dawid-Skene model to model the experts.
> > >
> > > Khetan & Oh 2016 and Shah et al. 2016 employ the generalized Dawid-Skene model, which considers the task difficulty, while we do not as we use the original Dawid-Skene model to model the experts. However, by employing the information from the datapoints, our results are robust to the correlated mistakes cases while they do not. We agree that combining the generalized Dawid-Skene model with our Max-MIG framework is an important future direction to explore. We will add this comparison, and the comparison with (3) in our revised version.
> > >
> > >
> > > Khetan, Ashish, and Sewoong Oh. "Achieving budget-optimality with adaptive schemes in crowdsourcing." Advances in Neural Information Processing Systems. 2016.
> > >
> > > Shah, Nihar B., Sivaraman Balakrishnan, and Martin J. Wainwright. "A permutation-based model for crowd labeling: Optimal estimation and robustness." arXiv preprint arXiv:1606.09632 (2016).

---

### Official Review · AnonReviewer2 · 2018-11-02
**More details on actual learning are required**

**Rating:** 6
**Confidence:** 4

**Review:**

Update after feedback: I would like to thank the authors for their detailed answers, it would be great to see some revisions in the paper also though (except new experimental results).
Especially thank you for providing details of a training procedure which I was missing in the initial draft. I hope to see them in the paper (at least some of them).

I have increased the rating to 6. Given new experimental results both on real data and forecaster comparison I would like to increase the rating to 7. However, I am not sure that this is fair to other authors who would might not be physically able to provide new experimental results due to computational constraints, please note that the experiments in this paper are rather 'light' in the standards of modern deep learning experiments and can be done within the rebuttal period.
====================================================


The paper finds a practical implementation of ideas from Kong & Schoenebeck (2018) for the learning with crowd problem. It proofs the claims from Kong & Schoenebeck (2018) for the specific family of data classifiers and crowd aggregators. From the general perspective, the papers proposes a method for joint training a classifier and a crowd label aggregator with particular consideration of correlated crowd labels.

The paper is fairly well-written and well-balanced between theoretical and empirical justification of the method. I see 1 major and 1 big issues with the paper.

Major issue: I am missing details of the actual procedure of training the model. Is MIG set as a loss function for the data classifier NN? Is crowd aggregator trained also as an NN with MIG as a loss function? How do the authors find the optimal p? Also, in order all the provided theory to work all the found data classifier NN, the aggregator and p should be exact maximisers of MIG as far as I understand. How do the author ensure that they find the exact maximisers? Also related to understanding how training works: on p.15 the authors claim “Note that our method can handle this simple correlated mistakes case and will give all useless experts weight zero based on Theorem 3.4.” I have trouble understanding why the proposed method should find these zero weights rather than it is just able to find them?

I am willing to change my judgement if the authors provide convincing details on the training procedure.

Big issue: Experimental settings.
a) Though it is interesting to see the analysis of the method under controlled environments of synthetic crowd labels with different properties that show benefits of the proposed method (such as dealing with correlated crowd labels), it would be also appealing to see the results with real labels, for example, Rodrigues & Pereira (2017) provide Amazon MTurk crowd labels for the LabelMe data
b) Is the proposed data-crowd forecaster the only method that uses crowd labels on the test data? While it can be argued that it is not straightforward in the test regime to include crowd labels into Crowd Layer, for example, without retraining the neural net, AggNet can use crowd labels without retraining the neural net part. In the presented format, it is unfair to compare the forecaster with the other methods because it uses more information, and essentially, the forecaster is not compared with anything (that uses the same information). It can be compared, at least, with pure Majority Voting, or more advanced pure crowdsourcing aggregation methods. Yes, they won’t use image data, but at least they can use the same amount of crowd label information, which would make a nice comparison with the presented related work and proposed NN: this is what you can get using just image data during test (Crowd Layer, Max-MIG, and others from the current paper), this is what you can get using just crowd labels during test (Majority Voting or, preferably, more advanced pure crowdsourcing aggregators), and this is what you can get using both image and crowd labels during test (the proposed forecaster and AggNet, for example)

Questions out of curiosity:
i). Does Max-MIG handle missing crowd labels for some data points? Did the author use missing labels in the experiments?
ii). Both the Dogs vs. Cats and CIFAR-10 datasets have more or less balanced data, i.e., the number of data points belonging to each ground truth class is similar between classes. Is this true for the LUNA16 dataset? If yes, have the authors tried their method with heavily imbalanced data? In my experience, some crowdsourcing methods may suffer with imbalanced data, for example, Crowd Layer does so on some data. This tendency of Crowd Layer is kind of confirmed on the provided Dogs vs. Cats in the naïve majority case, where based on crowd labels the first class dominates the second.

Other questions/issues/suggestions:
1. Until the formal introduction of the forecaster on page 4, it is not entirely clear what is the difference between the data classifier and data-crowd forecaster. It should be explained more clearly at the beginning that the 3 concepts (data classifier, crowd label aggregator and "data-crowd forecaster") are separated. Also some motivation why we should care about forecaster would be beneficial because one can argue that if we could train a NN that would make good enough predictions why we should waste resources on crowd labels. For example, the provided empirical results can be used as an argument for this.
2. From the introduction it is unclear that there are methods in crowdsourcing that do not rely on the assumption that data and crowd labels are independent given the ground truth labels. As mentioned in related works there are methods dealing with difficulty of data points, where models assume that crowd labels maybe biased on some data points due to their difficulty, e.g., if images are blurred, which violates this assumption.
Also the note that considering image difficulty violates the independence assumption could be added on page 3 around "[we] do not consider the image difficulty"
3. The beginning of page 4. I think it would be more clear to replace "5 experts' labels:" by $y^{[5]}=$
4. I suggest to move the caption of Figure 3 into the main text.
5. p.3 "However, these works are still not robust to correlated mistakes" - Why?
6. Data-crowds forecaster equation. It would be good to add some intuition about this choice. The product between the classifier and aggregator predictions seems reasonable, division on p_c is not that obvious. This expression presumably maximises the information gain introduced below. Some link between this equation and the gain introduction would be nice. Also, minor point – it is better to enlarge inner brackets ()_c
7. The formulation “To the best of our knowledge, our approach is a very early algorithm”, and namely “a very early algorithm” is unclear for me
8. Dual usage of “information intersection” as an assumption and as something that Max-MIG finds is confusing
9. Any comments how the learning rates were chosen are always beneficial
10. Proof of Proposition C.3: “Based on the result of Lemma C.2, by assuming that h ∗ ∈ H_{NN} , we can see (h ∗ , g∗ ,p ∗ ) is a maximizer of max_{h∈H_{NN} ,g∈G_{W A},p∈∆_C} MIGf (h, g,p)” – is expectation missing in the max equation? Is this shown below on page 13? If yes, then the authors should paraphrase this sentence as it does not imply that this is actually shown below
11. p.12 (and below) – what is $\mathbf{C}^m$? Is it $\mathbf{W}^m$?
12. p.15 (at the end of proof) $p \log q$ and $p \log p$ are not formally defined

Minor:
1. p.1 "of THE data-driven-based machine learning paradigm"
2. "crowds aggregator" -> "crowd aggregator"?
3. p.2 (and below) "between the data and crowdsourced labels i.e. the ground truth labelS"
4. Rodrigues & Pereira (2017) has a published version (AAAI) of their paper
5. p.2 "that model multiple experts individually and explicitly in A neural network"
6. p.3 "model the crowds by A Gaussian process"
7. p.3 "We model the crowds via confusion matriCES"
8. p.3 "only provide A theoretic framework and assume AN extremely high model complexity"
9. p.4 "forecast" for h and g -> "prediction"?
10. p.6 “between the data and the crowdsourced labelS”?
11. p.6 “However, in practice, with A finite number of datapoints”
12. p.6 “the experiment section will show that our picked H_{NN} and G_{W A} are sufficientLY simple to avoid over-fitting”
13. p.6 “We call them A Bayesian posterior data classifier / crowds aggregator / data-crowds forecaster, RESPECTEVILY”
14. p.6 “Theorem 3.4. With assumptionS 3.1, 3.3”
15. p.7 “DoctOr Net, the method proposed by Guan et al. (2017)”
16. p.7 “including the naive majority case since naive expert is independent with everything” – rephrasing is required, unclear what “independent with everything” means and who is “naïve expert”
17. Please capitalised names of conferences and journals in References
18. p.10 “she labels the image as “dog”/“cat” with THE probability 0.6/0.8 respectively”, “(e.g. B labels the image as “cat” with THE probability 0.5 and “dog” with THE probability 0.5 when the image has cats or dogs)”
19. p.12 “Lemma C.2. (Kong & Schoenebeck, 2018) With assumptionS 3.1, 3.3”, “Proposition C.3. [Independent mistakes] With assumptionS 3.1, 3.3”

---

> ### Author Response · Authors · 2018-11-13
> **Thank you for your careful review.**
>
> Thanks for your comments and questions. We will release our code after review.
>
> Q: >>Is MIG set as a loss function for the data classifier NN? Is crowd aggregator trained also as an NN with MIG as a loss function?
> A: <<We train the data classifier and the crowd aggregator together using -MIG(data classifier, crowd aggregator,p) as the loss function, i.e. they share the loss function.
>
> Q: >>How do the authors find the optimal p?
> A: <<We tune p as a hyperparameter and to maximize MIG(data classifier, crowd aggregator,p).
>
> Q: >>Also, in order all the provided theory to work all the found data classifier NN, the aggregator and p should be exact maximisers of MIG as far as I understand. How do the author ensure that they find the exact maximisers?
> A: <<We are not sure we understand this question. If this question asks about the robustness our algorithm, then our empirical results show that our algorithm is robust.
>
>
> Q: >>Also related to understanding how training works: on p.15 the authors claim “Note that our method can handle this simple correlated mistakes case and will give all useless experts weight zero based on Theorem 3.4.” I have trouble understanding why the proposed method should find these zero weights rather than it is just able to find them?
> A: <<Theoretically, our method should and is able to give the useless experts weight zero. In the simple correlated mistakes case, the best crowd aggregator gives the only useful expert all weight and other useless experts zero weights. During the training process, in order to maximize the mutual information between the classifier and aggregator, the SGD process of our algorithm will increase the weights of the useful experts and decrease the weights of the useless experts to a relatively small number, such that the trained aggregator approximates the best crowd aggregator. We will clarify it in our final version.
>
> Q: >>Real crowdsourced data
> A: <<Thanks for your suggestion. See our top comments.
>
> Q: >>Compare our data-crowd forecaster with AggNet
> A: <<Thanks for your suggestion. We compared our data-crowd forecaster with AggNet. The results still match our theory. When there are no correlated mistakes, we outperform AggNet or have very similar performances. When there are correlated mistakes, we outperform AggNet a lot (e.g. +30%). We have revised our paper and added this result.
>
> Q: >>Compare our crowd aggregator with pure crowdsourcing methods (Majority Voting or, preferably, more advanced pure crowdsourcing aggregators)
> A: <<This is still an unfair comparison. Although our crowd aggregator only takes the crowdsourced labels as input, the training process of our crowd aggregator incorporated the information from the images.
>
> Q: >>Does Max-MIG handle missing crowd labels for some data points? Did the author use missing labels in the experiments?
> A: <<The LabelMe data is in the missing label setting, the empirical results (our top comments) show that our algorithm handles this setting.
>
> Q: >> Both the Dogs vs. Cats and CIFAR-10 datasets have more or less balanced data, i.e., the number of data points belonging to each ground truth class is similar between classes. Is this true for the LUNA16 dataset? If yes, have the authors tried their method with heavily imbalanced data? In my experience, some crowdsourcing methods may suffer with imbalanced data, for example, Crowd Layer does so on some data. This tendency of Crowd Layer is kind of confirmed on the provided Dogs vs. Cats in the naïve majority case, where based on crowd labels the first class dominates the second.
> A: <<LUNA16 is highly imbalanced (85%, 15%). We will clarify it in our final version.
>
> We thank you for your careful review and will follow your suggestions on our writings and fix the typos.

---

> ### Author Response · Authors · 2018-11-22
> **The training procedure of Max-MIG**
>
> The training procedure of Max-MIG are illustrated below.
>
> Denote data classifier as h and crowds aggregator as g. The parameters of h (resp. g) is \theta_h (resp. \theta_g); the learning rate of h(resp. g) is \alpha_h (resp. \alpha_g). The crowdsourced dataset is denoted as D. (X, Y^{M}) pair denotes a batch of images and their corresponding crowdsourced labels Y^{M} from M experts. We tune the prior p as a hyperparameter.
>
> The implementation details, such as batch size, learning rate and network architecture for different datasets, are illustrated in Page 12-13 in our paper.
>
> Step 1: Initialization for the experts' parameters in the crowds aggregator   (Please refer to Page 13 in our paper for more details)
>
> Step 2 :
>
>     For t in 1, 2, ..., T
>
> 	    sample mini-batch (X, Y^{M}))  from D
>
> 	    left_output = h(X)
>
> 	    right_output= g(Y^{M})
>
> 	    Loss = - MIG(left_output,right_output,p)   (Please refer to Page 5 in our paper for more details)
>
> 	    \theta_h = \theta_h - \alpha_h * \nabla{\theta_h}Loss
>
> 	    \theta_g = \theta_g - \alpha_g * \nabla{\theta_g}Loss

---

### Official Review · AnonReviewer3 · 2018-11-05
**Generally well-written paper on a new information theoretical approach for learning from crowdsourced data.**

**Rating:** 6
**Confidence:** 4

**Review:**

Top pros:
- Well motivated approach with good examples from clinical setting
- Sound proof on why information theoretical approach is better than MLE based approaches
- Experiments on diversified data sets to show their approach's performance, with good implementation details.

Top cons:
- Fairly strong assumption on the existence of mutually independent senior experts in the labeling process
- Hard-to-check assumption for Theorem 3.4 for real world problems, on the sufficiency of senior expert's info to predict the true class label

The paper is in general well written, and builds upon existing work on crowdsourced data mining and co-training. I believe this line of work will benefit the community in taking a more information theoretical approach with relaxed assumptions on the data collection process. My main feedback is how to check the existence of senior experts in real-world applications. In particular,
- If the labels are collected from an unknown setup (e.g. on AMT), where it is hard to establish the dependency structure of the experts, how can we use such approaches effectively?
- Even if there exists a clear line between senior/junior experts in the labeling process, how do we know or check that the senior experts' opinion can sufficiently estimate the true labels?

In the experiment section, the label data was collected with a build-in assumption of senior/junior labelers, and we also know exactly who are senior/junior experts. So it is not surprising that the proposed approach outperforms other approaches. It's also interesting to see that AggNet isn't that bad in general compared to the proposed approach (except on LUNA16). What if we combine all experts in one setting and apply the proposed approach without prior knowledge of who are senior/junior? Also, did you require all experts to label ALL the data points or only a subset of training data points?

Minor points:
- I don't believe "Naive majority" is an interesting setting - we can easily detect those junior experts that always label cases with one class, and remove these experts from the system, in practice.
- I wouldn't call this an "early" algorithm as it indicates it's somewhat pre-mature. Just call this a novel approach that is in the early phase, and more sophisticated approach can be further developed.

---

> ### Author Response · Authors · 2018-11-13
> **Our algorithm learns from crowds without knowing the information structure among the crowds a priori.**
>
> Thanks for your comments and questions. There might be some misunderstanding and we want to clarify it here: our algorithm is not ad hoc and it is independent of any prior knowledge about the information structures and identities of the senior/junior expert. i.e. our algorithm learns from crowds without knowing the information structure among the crowds a priori.
>
> Q: >>If the labels are collected from an unknown setup (e.g. on AMT), where it is hard to establish the dependency structure of the experts, how can we use such approaches effectively?
> >>So it is not surprising that the proposed approach outperforms other approaches. It's also interesting to see that AggNet isn't that bad in general compared to the proposed approach (except on LUNA16). What if we combine all experts in one setting and apply the proposed approach without prior knowledge of who are senior/junior?
>
> A: <<Our algorithm does not need to know the dependency structure of the experts nor the identity of the senior or junior experts. In detail, our algorithm's input is (datapoints and crowdsourced labels) and the initialization is also independent of the dependency structure of the experts nor the identity of the senior or junior experts.
>
> Q: >>(Top cons)Hard-to-check assumption for Theorem 3.4 for real-world problems, on the sufficiency of senior expert's info to predict the true class label
> >>Even if there exists a clear line between senior/junior experts in the labeling process, how do we know or check that the senior experts' opinion can sufficiently estimate the true labels?
> A: <<To implement our algorithm, we also do not need to check the sufficient statistic assumption.
>
> Q: >>(Top cons)Fairly strong assumption on the existence of mutually independent senior experts in the labeling process
> A: <<1)The general MAX-MIG framework does not need this assumption and the assumption can be relaxed by providing a more complicated aggregator model (see the last paragraph of the conclusion section). We agree that this can be an interesting direction to explore in future work. 2) Our results on real data show that our current implementation of MAX-MIG, with weighted average aggregator model, is still robust to the real situation empirically (see our top comments).
>
> Q: >>Did you require all experts to label ALL the data points or only a subset of training data points?
> A: <<The LabelMe data is in the missing label setting, the empirical results (our top comments) show that our algorithm handles this setting.
>
> Q: >>I don't believe "Naive majority" is an interesting setting - we can easily detect those junior experts that always label cases with one class, and remove these experts from the system, in practice.
> A: <<In our experiments, we make the naive majority always label 1 to show that other methods (e.g. majority vote) cannot handle this setting. However, in fact, if the naive majority label 1 with prob 0.9 and 0 with prob 0.1, we cannot easily remove them and other methods still cannot handle this setting, while our algorithm will not be affected by these kinds of naive majority based on our theory.
>
> Q: >>The term ``early''
> A: <<We will revise this term.

---

### Author Response · Authors · 2018-11-13
**Our method achieves the state-of-the-art results on real-world dataset, LabelMe**

To all reviewers:

Thank you all for suggesting experiments on real Amazon MTurk data. We follow the second reviewer's suggestion and run our algorithm using LabelMe data, which has the real Amazon MTurk crowd labels (it is also the missing label setting with 59 annotators and each image was labeled by an average of 2.547 workers), from Rodrigues and Pereira (2017). We also achieve the state of art in this real data case. Here is the result: Max-Mig 86.42 +/- 0.36, Majority vote 80.41 +/- 0.56, Crowd Layer 83.65 +/- 0.50, Doctor Net 80.56 +/- 0.59, AggNet 85.20 +/- 0.26.

We have revised our paper and added this result. We will also release our code after review.

---

### Meta-Review · Area_Chair1 · 2018-12-17
**Interesting idea for interpreting crowd-sourced labels**

**Confidence:** 4
**Recommendation:** Accept (Poster)

**Metareview:**

This paper proposes an interesting approach to leveraging crowd-sourced labels, along with an ML model learned from the data itself.

The reviewers were unanimous in their vote to accept.